# Contributing to Biochemistry and Optoelectronics: Pyrrolo[1′,2′:2,3]imidazo[1,5-*a*]indoles and Cyclohepta[4,5]pyrrolo[1,2-*c*]pyrrolo[1,2-*a*]imidazoles via [3+2] Annulation of Acylethynylcycloalka[*b*]pyrroles with Δ^1^-Pyrrolines

**DOI:** 10.3390/ijms24043404

**Published:** 2023-02-08

**Authors:** Ludmila A. Oparina, Nikita A. Kolyvanov, Igor A. Ushakov, Lina P. Nikitina, Olga V. Petrova, Lyubov N. Sobenina, Konstantin B. Petrushenko, Boris A. Trofimov

**Affiliations:** A. E. Favorsky Irkutsk Institute of Chemistry, Siberian Division of the Russian Academy of Sciences, 1 Favorsky Str., Irkutsk 664033, Russia

**Keywords:** acylethynylcycloalka[*b*]pyrroles, Δ^1^-pyrrolines, [3+2] annulation, pyrrolo[1′,2′:2,3]imidazo[1,5-*a*]indoles, cyclohepta[4,5]pyrrolo[1,2-*c*]pyrrolo[1,2-*a*]imidazoles

## Abstract

Available pyrrolylalkynones with tetrahydroindolyl, cycloalkanopyrrolyl, and dihydrobenzo[*g*]indolyl moieties, acylethynylcycloalka[*b*]pyrroles, are readily annulated with Δ^1^-pyrrolines (MeCN/THF, 70 °C, 8 h) to afford a series of novel pyrrolo[1′,2′:2,3]imidazo[1,5-*a*]indoles and cyclohepta[4,5]pyrrolo[1,2-*c*]pyrrolo[1,2-*a*]imidazoles functionalized with an acylethenyl group in up to an 81% yield. This original synthetic approach contributes to the arsenal of chemical methods promoting drug discovery. Photophysical studies show that some of the synthesized compounds, e.g., benzo[*g*]pyrroloimidazoindoles, are prospective candidates for TADF emitters of OLED.

## 1. Introduction

Nitrogen-fused heterocycles in general and pyrroloimidazoles in particular are important structural motifs frequently found in a number of natural products and bioactive molecules [1,2,3,4,5,6,7,8,9,10,11,12] (Figure 1), as well as contained in key components for optoelectronic devices [13,14,15]. Such compounds act as extracellular signal-regulated kinase (ERK) inhibitors [6], anticancer drugs [7,8], nervous depressants and analgesics [9], antibiotics [10], inhibitors of glucosamine deacetylase (LpxC) [11], neuropeptide S receptor (NPSR) antagonists [12], blue-emitting luminophores [13], TADF (thermally activated delayed fluorescence) emitters [14], and compounds with cell imaging properties [15].

The known approaches to the synthesis of pyrrolo[1,2-*c*]imidazoles include the intermolecular cyclocondensation of pyrrole, isocyanate and phosgene or thiophosgene [16,17], benzotriazol-1-yl(1*H*-pyrrol-2-yl)methanone with ketone, isocyanate or isothiocyanate [18]; pyrrole-2-carboxylic acid, amine and carbonyl diimidazole [5,19], Rh(III)- and Pd(0)-catalyzed C2-H functionalization of pyrroles with alkynes, alkenes and diazo compounds [20,21]; post-Ugi cascade reaction [7]; and flash vacuum pyrolysis of imidazol-1-ylacrylates [22].

Recently [23], we have serendipitously found that as a convenient platform for the construction of a dipyrrolo[1,2-*a*:1′,2′-*c*]imidazole core, acylethynylpyrroles, readily available from the Al_2_O_3_-promoted cross-coupling of acylbromoacetylenes with pyrroles [24], can be successfully employed. The similar condensed heterocyclic systems are of high medicinal relevance as they are the principal structural units of compounds with antiviral properties [1,2,3]. As a further development of this original catalyst-free [3+2] annulation, here we disclose a convenient synthesis of pyrrolo[1′,2′:2,3]imidazo[1,5-*a*]indoles and cyclohepta[4,5]pyrrolo[1,2-*c*]pyrrolo[1,2-*a*]imidazoles, so far unknown biochemically related polycondensed heterocyclic systems comprising such life-sustaining motifs as indole and pyrrole, from Δ^1^-pyrrolines and acylethynylpyrroles fused with cycloalkyl groups, namely acylethynyltetrahydroindoles **1**, -dihydrobenzo[*g*]indole **2**, and cyclohepta[*b*]pyrrole **3**.

## 2. Results and Discussion

As a model for primary optimization, the reaction of benzoylethynyltetrahydroindole **1a** with pyrroline **4a**, conveniently handled as its trimer, has been chosen (Table 1).

Although the reaction in MeOH was completed within 24 h at room temperature (starting **1a** was entirely consumed), the yield of product **5a** did not exceed 11% (entry 1). In this case, the major products were acetyltetrahydroindole and methyl benzoate (60% and 72% yields). This result may be rationalized by a competitive nucleophilic addition of MeOH to **1a** catalyzed by pyrroline **4a** as a strong base. Nucleophilic addition of water to adduct **A** and the subsequent decomposition of the intermediate **B** gives diketone **C**, which adds a second molecule of MeOH and the intermediate **D** thus formed decomposes to acetyltetrahydroindole and methyl benzoate (Figure 1).

In MeCN (rt, 24 h), the reaction results in 12% yield of adduct **5a** (entry 2) that can be attributed to the limited solubility of the starting materials in the reaction medium. Solvent screening revealed that a switch to MeCN/THF (1:1, *v*/*v*) gave better results (entries 2–4). Only a 1:1 ratio of MeCN/THF provided a homogenic reaction mixture. Therefore, other solvent ratios were not employed. After the short optimization of the reaction conditions, we found that, upon heating (70 °C), the comparable yields (80–81%, Table 1, entries 5, 6) were achieved in a shorter time than at room temperature (8 h vs. 72 h). The reaction conditions do not noticeably influence the ratio of *E*/*Z* isomers (~4:1). This prompted us to study the reaction of tetrahydroindole **1** with pyrrolines **4b**–**e** under similar conditions (MeCN/THF, 70 °C, 8 h).

The experiments revealed that the reaction of tetrahydroindole **1a** with pyrrolines **4b**–**d** bearing the C2-alkyl group (Me, Et, Pr*^i^*) under these conditions proceeded efficiently to afford pyrrolo[1′,2′:2,3]imidazo[1,5-*a*]indoles **5b**–**d** in good to high yields (Figure 2). There is a clear influence of the steric factors of the substituent at position 2 of pyrroline **4** on the yield of product **5**: the more sterically compact methyl group compared to isopropyl group gives the product in a higher yield (79% vs. 60%). The presence of a *tert*-butyl group near the reaction center creates steric hindrance, preventing a more favorable orientation of the reagents during the reaction. In this case, just a trace amount of the target product **5e** was detectable (^1^H NMR spectrum) in the reaction mixture. In addition, the negative effect of substituents in the pyrroline ring on the rate of cycloaddition and product yields is associated not only with steric hindrance, but also with the electron-donating action of alkyl substituents, which stabilizes the emerging positive charge in position 2 of the pyrroline ring. At the same time, the electron-withdrawing substituent at the triple bond of tetrahydroindole **1** had no noticeable effect: the reaction of benzoyl- **1a** and thenoyl- **1b** derivatives with 2-methylpyrroline **4b** proceeded under these conditions with comparable efficiency (79% and 72% yield). The synthesized pyrrolo[1′,2′:2,3]imidazo[1,5-*a*]indoles **5a**–**f** were isolated as a mixture of the *E*- and *Z*-isomers in a 4:1 ratio.

The reaction of dihydrobenzo[*g*]indole **2** with pyrroline **4** was expectedly somewhat more reluctant than that with tetrahydroindole **1** (Figure 3). Under analogous conditions, unsubstituted pyrroline **4a** provided the corresponding adduct **6a** in a 70% yield (conversion of **2** was 88%). In contrast to the similar reaction with tetrahydroindole (see above), 2-alkylpyrrolines **4b**–**d** led to the formation of pyrroloimidazoles **6b**–**d** only in a 26–44% yield (vs. 60–79% with tetrahydroindole) (conversion of **2** was 35–56%). Pyrroline **4e** with *tert*-butyl at the position 2 gave no even traces of the expected annulation product. An obvious cause of the reaction inhibition is steric repulsion between the benzene ring annulated with the tetrahydroindole moiety and the C2 substituent in pyrroline **4e**.

A similar effect of the C2 substituent on annulation was observed in the reaction of benzoylethynylcyclohepta[*b*]pyrrole **3** with pyrroline **4** (Figure 4). Pyrrole **3** reacted with pyrroline **4a** to form the expected tetracyclic adduct **7a** in a 76% yield as an *E*/*Z* isomer (4:1). In Me-pyrroline **4b**, the yield of adduct **7b** slightly decreased (conversion of the starting pyrrole **3** was 82% vs. 90% with pyrroline **4a**). The replacement of the methyl group by ethyl or isopropyl ones in pyrroline **4** reduced the product yields to 32–34%. This is probably due to the steric effect of the fused cycloheptyl moiety with a different conformation to cyclohexyl.

The structures of dipyrroloimidazoles **5**–**7** were proved by NMR (^1^H, ^13^C, ^15^N, including 2D correlations), IR spectroscopy, and mass spectrometry (See Appendix A).

The above experimental data evidence that, like in the previous work [23], the nucleophilic attack of the pyrroline nitrogen at the triple bond plays a key role here. The proton transfer from the pyrrole NH moiety to the vinyl carbanionic center in intermediate **A** delivers pyrrolate-centered intermediate **B**. The free rotation along the C−C bond allows a conformation suitable for the formation of a C2−N covalent bond to afford a pyrroloimidazole scaffold of products **5**–**7** (Figure 5, on the example of products **5**).

In a number of publications ([25] and ref. cite therein) concerning the reactions of nitrogen nucleophiles (pyridines, quinolones, indolizines, etc.) with activated alkynes, the reversible formation of 1,3(4)-dipolar complexes (zwitterions) as key intermediates was proved to occur. As follows from Figure 5, this step is implied to be a major one in the reaction studied and therefore its mechanism does not likely differ much in other details from the commonly accepted ones.

As suggested in article [23], the concerted [3+2] cycloaddition is also not excluded.

Benzo[*g*]indole scaffold is used in the synthesis of fluorescent organic molecules that have wide biomedical and technical applications [26,27]. Our interest in the photophysics of the synthesized benzo[*g*]pyrroloimidazoindoles **6** is instigated by the presence of intense long-wavelength (*λ*_abs_(max)~410 nm, *ε* ≈ 40,000 M^−1^ cm^−1^) electronic absorption bands with maxima in the violet region and blue photoluminescence, which could make them attractive materials in optoelectronics for blue organic light-emitting diodes (OLEDs). It is clear that the optoelectronics-related properties of benzo[*g*]pyrroloimidazoindoles **6a**–**d** are associated with the benzo[*g*]indole fragment in their structure, as most conjugated and enriched p-electrons. Since compounds **6a**–**d** differ from each other only by alkyl substituent, their photophysical properties should not be noticeably different. In this work, preliminary experimental and theoretical studies of the spectral and photophysical properties were performed using (*E*)-2-(5,6,10,11,12,12*a*-hexahydro-8*H*-benzo[*g*]pyrrolo[2’,1’:2,3]imidazo[1,5-*a*]indol-8-ylidene)-1-phenylethan-1-one (**6a**) as an example.

In liquid media, despite the ππ*-character and strong oscillator strength of the fluorescent S_1_ state (f = 1.09), as well as the favorable relative position of the forbidden state S_2_ (n_O_π*+ππ*, f = 0.04), the fluorescence quantum yield (Φ*_F_*) of the synthesized compound turned out to be low (Table 2 and Table 3, Figure 2). This is most likely explained by the effective intersystem crossing (ISC) between the fluorescent S_1_ (ππ*_ST_, 3.54 eV) and the closest to it lower energy triplet state T_4_, (n_O_π*+ππ*, 3.43 eV) with different orbital types according to El Sayed’s rules [28] (Table 3, Figure 2).

On the other hand, it is known that in solid media organic molecules with a small singlet–triplet (S-T) gap and a donor–acceptor (D-A) character can exhibit thermally activated delayed fluorescence (TADF) [29,30]. The mechanism is based on the thermal upconversion of the triplet excitons into singlets via the reverse intersystem crossing. TADF has gained considerable interest in recent years, since the materials exhibiting TADF can act as high-performance emitters in OLEDs [31,32]. Since quantum chemical calculations (Table 3) predict that benzo[*g*]pyrroloimidazoindoles have small S-T splitting (Δ*E*_ST_ = 0.11 eV for **6a**), as well as the fact that they are D-A type compounds (donor–pyrrole ring and acceptor–benzoyl fragment), then according to the literature criteria, these compounds can be good candidates for TADF emitters for OLEDs. In this regard, in the near future we plan to carry out spectral-luminescence and kinetic studies of benzo[*g*]pyrroloimidazoindoles in solid media.

## 3. Materials and Methods

### 3.1. General Information

NMR spectra were recorded from solutions in CDCl_3_ on Bruker DPX-400 and AV-400 spectrometers (Bruker, Billerica, MA, USA) (400.1 MHz for ^1^H, 100.6 MHz for ^13^C, and 40.5 MHz for ^15^N). Chemical shifts (*δ*) were quoted in parts per million (ppm). The residual solvent peak, *δ*_H_ 7.27 and *δ*_C_ 77.10, was used as a reference. Coupling constants (*J*) were reported in hertz (Hz). The following abbreviations were used to express the multiplicates: s (singlet), d (doublet), dd (doublet of doublet), t (triplet), dt (doublet of triplet), dq (doublet of quartet), m (multiplet), nr (narrow), br (broad). The ^15^N chemical shifts were referenced to CH_3_NO_2_. The configurational assignment and the substituent location for the compounds **5**–**7** are based on 2D (NOESY, ^1^Н-^13^С HSQC, ^1^Н-^13^С HMBC, ^1^Н-^15^N HMBC) NMR spectroscopy data.

UV/Vis absorption spectra were measured on a Lambda-35 (Perkin-Elmer, Waltham, MA, USA) spectrophotometer. Fluorescence spectra were measured on a FLSP-920 combined steady-state and time-resolved fluorescence spectrometer (Edinburgh Instrument, Livingston, UK). All the solvents employed for the spectroscopic measurements were of UV spectroscopic grade (Merck, Rahway, NJ, USA). For fluorescence measurements, dilute solutions with an absorbance of less than 0.1 (at 1 cm optical path length) at the absorption maximum were used. The fluorescence measurements were performed with a 90° standard geometry. The fluorescence quantum yields (Φ*_F_*) of **6a** were evaluated relative to anthracene (Φ*_F_* = 0.27 in EtOH as the reference and was corrected from the dependence of the refractive index of the solvent [33]). The temperature for fluorescence measurements was kept constant at 298 K. All quantum chemistry calculations were carried out using the Gaussian 09.B.01 program package [34]. The ground-state (S_0_) geometry optimizations and the vertical excitations S_0_ → S_i_ (i = 1–5) at the S_0_ geometry were calculated with the TD-CAMB3LYP method, with a split valence with polarization SVP basis set. An analysis of the nature of the transitions, fraction of electron charge transferred (*q_CT_*), distance of charge transfer (*D_CT_*), and dipole moment variation at excitation (Δ*μ*) was carried out using Multiwfn 3.8. software [35].

IR spectra were obtained on a Varian 3100 IF-IR spectrometer (Digilab LLC, USA) (400–4000 cm^−1^) as thin films dispersed from CDCl_3_. Mass spectra of synthesized compounds were recorded on a GCMS-QP5050A spectrometer from Shimadzu Company. High-resolution mass spectral analyses were performed from acetonitrile solution with 0.1% HFBA on an HPLC Agilent 1200/Agilent 6210 TOF instrument equipped with an electrospray ionization (ESI) source (Agilent, USA). Melting points (uncorrected) were measured on a melting point apparatus SGW-X-4 (China). Thin-layer chromatography was carried out on Merck silica gel 60 F254 pre-coated aluminum foil sheets which were visualized using UV light (254 nm). Column chromatography was carried out using slurry-packed Alfa Aesar silica gel (SiO_2_), 70–230 mesh, pore size 60 Å.

### 3.2. Preparation and Characterization of Substrates

Pyrrolines **4a** [36] and **4c**–**e** [37] were prepared by following the same procedure as described in the literature. 2-Methyl-1-pyrroline **4b** was purchased from commercial sources (Alfa Aesar, Haverhill, MA, USA). Starting acylethynyltetrahydroindole **1a**,**b** and acylethynyldihydrobenzo[*g*]indole **2** were obtained from corresponding pyrroles and acylbromoacetylenes in the presence of Al_2_O_3_ according to published methods [24,38]. 3-(1,4,5,6,7,8-Hexahydrocyclohepta[*b*]pyrrol-2-yl)-1-phenylprop-2-yn-1-one **3** was prepared for the first time from 1,4,5,6,7,8-hexahydrocyclohepta[*b*]pyrrole [39] and benzoylbromoacetylene.

1,4,5,6,7,8-Hexahydrocyclohepta[*b*]pyrrole (0.676 g, 5.0 mmol) and 1-benzoyl-2-bromoacetylene (1.045 g, 5.0 mmol) were ground together at rt with 17.0 g (10-fold amount) of Al_2_O_3_ (chromatography grade, Merck, pH 6.8–7.8) in a porcelain mortar for 1–2 min. The reaction mixture self-heated (5–8 °C) and within 10 min turned from yellow to orange-brown. After 3 h, the reaction products were extracted sequentially with hexane (50 mL), *n*-hexane–Et_2_O (2:1–1:2) (150 mL), and Et_2_O (100 mL). The fractions were further chromatographed on a column (Al_2_O_3_) to yield ethynylpyrrole **3**.

3-(1,4,5,6,7,8-Hexahydrocyclohepta[*b*]pyrrol-2-yl)-1-phenylprop-2-yn-1-one (**3**). Yellow crystals, 0.986 g, 75%, mp 131–132 °C.

^1^H NMR (400.1 MHz, CDCl_3_) δ: 8.54 (br s, 1H, NH), 8.16–8.14 (m, 2H, H*_o_*, Ph), 7.60–7.56 (m, 1H, H*_p_*, Ph), 7.50–7.46 (m, 2H, H*_m_*, Ph), 6.65 (d, *J* = 2.4 Hz, 1H, H-3, pyrrole), 2.72–2.70 (m, 2H, CH_2_), 2.58–2.56 (m, 2H, CH_2_), 1.86–1.80 (m, 2H, CH_2_), 1.72–1.64 (m, 4H, 2CH_2_).

^13^C NMR (100.6 MHz, CDCl_3_): 177.7, 139.2, 139.2, 137.2, 133.5, 129.3 (2С), 128.5 (2С), 125.2, 122.6, 105.4, 93.7, 32.9, 29.5, 28.9, 28.2, 27.4.

IR (KBr): *ν =* 3285 (NH), 2152 (C≡C), 1613 (C=O).

Found: C, 81.98%; H, 6.53%; N, 5.28%. Calcd. for C_18_H_17_NO: C, 82.10%; H, 6.51%; N, 5.32%.

### 3.3. General Procedure for the Synthesis of Compounds ***5***–***7***

To a solution of ethynylpyrrole **1**–**3** (0.5 mmol) in the mixed solvent of MeCN (0.5 mL) and THF (0.5 mL), pyrroline **4** (0.5 mmol) was added and the resulting mixture was stirred at 70 °C in an oil bath for 8 h. Then, the solvents were removed under reduced pressure, and residue was purified by column chromatography (SiO_2_, eluent: hexane/Et_2_O 1:1, *v*/*v*) to give adducts **5**–**7**.

2-(1,2,3,3a,5,6,7,8-Octahydro-10*H*-pyrrolo[1′,2′:2,3]imidazo[1,5-*a*]indol-10-ylidene)-1-phenylethan-1-one (**5a**). Yield 129 mg, 81%. E/Z-ratio ~4:1. Yellow solid, mp 172–174 °C (hexane).

^1^Н NMR (400.1 MHz, CDCl_3_): 7.98–7.96 (m, 2H, H*_o_*, Ph), 7.48 (s, 1H, H-9), 7.46–7.39 (m, 3H, H*_m_*_,*p*_, Ph), 5.91 (s, 1H, =CH), 5.43 (dd, *J* = 8.3, 5.6 Hz, 1H, H-3a), 3.61–3.56 (m, 1H, H-1), 3.36 (dt, *J* = 9.7, 7.9 Hz, 1H, H-1), 2.64–2.56 (m, 4H, H-5, H-8), 2.39–2.22 (m, 3H, H-2, H-3), 1.88–1.73 (m, 4H, H-6, H-7), 1.58–1.48 (m, 1H, H-3)—*E*-isomer; 7.98–7.96 (m, 2H, H*_o_*, Ph), 7.46–7.39 (m, 3H, H*_m_*_,*p*_, Ph), 6.21 (s, 1H, H-9), 6.20 (s, 1H, =CH), 5.61 (dd, *J* = 7.8, 6.0 Hz, 1H, H-3a), 4.32–4.27 (m, 1H, H-1), 3.10 (dt, *J* = 10.1, 8.6 Hz, 1H, H-1), 2.66–2.54 (m, 4H, H-5, H-8), 2.39–2.22 (m, 3H, H-2, H-3), 1.88–1.73 (m, 4H, H-6, H-7), 1.58–1.58 (m, 1H, H-3)—*Z*-isomer.

^13^С{^1^H} NMR (100.6 MHz, CDCl_3_): 187.2 (C=O), 155.0 (C-10), 141.7 (C*_i_*, Ph), 130.9 (C-9a), 130.6 (C*_p_*, Ph), 128.7 (C-4a), 128.2 (C*_m_*, Ph), 127.5 (C*_o_*, Ph), 124.6 (C-8a), 111.3 (C-9), 90.3 (=CH), 76.9 (C-3a), 48.5 (C-1), 30.5 (C-3), 26.8 (C-2), 23.5, 23.4, 22.9, 22.3 (C-5–8)—*E*-isomer; 186.5 (C=O), 153.6 (C-10), 141.4 (C*_i_*, Ph), 132.9 (C-9a), 130.8 (C*_p_*, Ph), 128.4 (C-4a), 128.2 (C*_m_*, Ph), 127.6 (C*_o_*, Ph), 125.0 (C-8a), 101.6 (C-9), 88.1 (=CH), 79.5 (C-3a), 51.4 (C-1), 31.4 (C-3), 26.9 (C-2), 23.5, 23.4, 22.8, 22.0 (C-5–8)—*Z*-isomer.

^15^N NMR (40.6 MHz, CDCl_3_): −265.7 (N-11), −206.3 (N-4)—*E*-isomer; −257.6 (N-11), −206.3 (N-4)—*Z*-isomer.

IR (film): *ν* = 1631 (C=O), 1578 (C=C) cm^−1^.

MS (EI): *m/z* (%) = 318 (M^+^, 47%), 213 (55), 146 (14), 145 (13), 105 (PhCO^+^, 100), 77 (Ph^+^, 90), 51 (22), 44 (11), 41 (23).

HRMS (ESI-TOF): found 319.1811. Calcd. for [C_21_H_22_N_2_O+H]^+^ 319.1810.

2-(3a-Methyl-1,2,3,3a,5,6,7,8-octahydro-10*H*-pyrrolo[1′,2′:2,3]imidazo[1,5-*a*]indol-10-ylidene)-1-phenylethan-1-one (**5b**). Yield 131 mg, 79%. E/Z-ratio ~4:1. Pale yellow solid, mp 149–152 °C (hexane).

^1^Н NMR (400.1 MHz, CDCl_3_): 7.98–7.95 (m, 2H, H*_o_*, Ph), 7.49 (s, 1H, H-9), 7.43–7.40 (m, 3H, H*_m_*_,*p*_, Ph), 5.88 (s, 1H, =CH), 3.64–3.59 (m, 1H, H-1), 3.45–3.38 (m, 1H, H-1), 2.76–2.58 (m, 4H, H-5, H-8), 2.34–2.20 (m, 3H, H-2, H-3), 1.95–1.92 (m, 1H, H-3), 1.82–1.74 (m, 4H, H-6, H-7), 1.61 (s, 3H, Me-3a)—*E*-isomer; 7.98–7.95 (m, 2H, H*_o_*, Ph), 7.43–7.40 (m, 3H, H*_m_*_,*p*_, Ph), 6.18 (s, 1H, H-9), 6.16 (s, 1H, =CH), 4.33–4.28 (m, 1H, H-1), 3.19–3.12 (m, 1H, H-1), 2.76–2.58 (m, 4H, H-5, H-8), 2.34–2.20 (m, 3H, H-2, H-3), 1.95–1.92 (m, 1H, H-3), 1.82–1.74 (m, 4H, H-6, H-7), 1.68 (s, 3H, Me-3a)—*Z*-isomer.

^13^С{^1^H} NMR (100.6 MHz, CDCl_3_): 187.2 (C=O), 154.5 (C-10), 141.8 (C*_i_*, Ph), 130.7 (C-9a), 130.5 (C*_p_*, Ph), 128.3 (C-4a), 128.1 (C*_m_*, Ph), 127.4 (C*_o_*, Ph), 124.8 (C-8a), 111.1 (C-9), 90.2 (=CH), 84.8 (C-3a), 48.4 (C-1), 35.2 (C-3), 26.9 (Me-3a), 25.6 (C-2), 23.5, 23.3, 23.1, 22.4 (C-5–8)—*E*-isomer; 186.2 (C=O), 153.2 (C-10), 141.6 (C*_i_*, Ph), 132.0 (C-9a), 130.5 (C*_p_*, Ph), 129.7 (C-4a), 128.3 (C*_m_*, Ph), 127.5 (C*_o_*, Ph), 125.1 (C-8a), 101.2 (C-9), 87.9 (C-3a), 87.7 (=CH), 51.3 (C-1), 36.2 (C-3), 27.0 (Me-3a), 25.8 (C-2), 23.6, 23.3, 23.0, 22.2 (C-5–8)—*Z*-isomer.

^15^N NMR (40.6 MHz, CDCl_3_): −255.1 (N-11), −194.4 (N-4)—*E*-isomer; −244.9 (N-11), −194.4 (N-4)—*Z*-isomer.

IR (film): *ν* = 1632 (C=O), 1578 (C=C).

MS (EI): *m/z* (%) = 332 (M^+^, 29%), 227 (100), 105 (PhCO^+^, 34), 77 (Ph^+^, 45).

HRMS (ESI-TOF): found 333.1968. Calcd. for [C_22_H_24_N_2_O+H]^+^ 333.1967.

2-(3a-Ethyl-1,2,3,3a,5,6,7,8-octahydro-10*H-*pyrrolo[1′,2′:2,3]imidazo[1,5-*a*]indol-10-ylidene)-1-phenylethan-1-one (**5c**). Yield 111 mg, 64%. E/Z-ratio ~4:1. Light yellow solid, mp 148–151 °C (hexane).

^1^Н NMR (400.1 MHz, CDCl_3_): 7.98–7.96 (m, 2H, H*_o_*, Ph), 7.47 (s, 1H, H-9), 7.43–7.41 (m, 3H, H*_m_*_,*p*_, Ph), 5.91 (s, 1H, =CH), 3.57–3.43 (m, 2H, H-1), 2.65–2.64 (m, 4H, H-5, H-8), 2.29–2.17 (m, 3H, H-2, H-3), 2.04–1.85 (m, 3H, H-3, C*H*_2_Me), 1.81–1.75 (m, 4H, H-6, H-7), 0.50 (t, *J* = 7.2 Hz, 3H, *Me*CH_2_)—*E*-isomer; 7.98–7.96 (m, 2H, H*_o_*, Ph), 7.43–7.41 (m, 3H, H*_m_*_,*p*_, Ph), 6.16 (s, 2H, H-9, =CH), 4.24–4.18 (m, 1H, H-1), 3.26–3.19 (m, 1H, H-1), 2.64–2.58 (m, 4H, H-5, H-8), 2.29–2.17 (m, 3H, H-2, H-3), 2.04–1.85 (m, 7H, H-3, H-6, H-7, C*H*_2_Me), 0.50 (t, *J* = 7.2 Hz, 3H, *Me*CH_2_)—*Z*-isomer.

^13^С{^1^H} NMR (100.6 MHz, CDCl_3_): 187.3 (C=O), 155.5 (C-10), 141.9 (C*_i_*, Ph), 131.2 (C-9a), 130.5 (C*_p_*, Ph), 128.5 (C-4a), 128.1 (C*_m_*, Ph), 127.5 (C*_o_*, Ph), 124.9 (C-8a), 110.9 (C-9), 90.2 (=CH), 87.9 (C-3a), 48.6 (C-1), 34.8 (Me*C*H_2_), 30.8 (C-3), 26.9 (C-2), 23.6, 23.5, 23.2, 22.5 (C-5–8), 6.6 (*Me*CH_2_)—*E*-isomer; 186.1 (C=O), 153.9 (C-10), 141.7 (C*_i_*, Ph), 133.5 (C-9a), 130.7 (C*_p_*, Ph), 128.5 (C-4a), 128.1 (C*_m_*, Ph), 127.6 (C*_o_*, Ph), 125.1 (C-8a), 100.9 (C-9), 90.9 (C-3a), 87.5 (=CH), 51.4 (C-1), 35.6 (*C*H_2_Me), 31.2 (C-3), 26.9 (C-2), 23.7, 23.4, 23.1, 22.3 (C-5–8), 6.6 (*Me*CH_2_)—*Z*-isomer.

^15^N NMR (40.6 MHz, CDCl_3_): −259.8 (N-11), −199.7 (N-4)—*E*-isomer; −251.0 (N-11), −199.7 (N-4)—*Z*-isomer.

IR (film): *ν* = 1630 (C=O), 1578 (C=C).

MS (EI): *m/z* (%) = 346 (M^+^, 43%), 331 (33), 242 (19), 241 (100), 105 (PhCO^+^, 32), 77 (Ph^+^, 55), 41 (15).

HRMS (ESI-TOF): found 347.2119. Calcd. for [C_23_H_26_N_2_O+H]^+^ 347.2123.

2-(3a-Isopropyl-1,2,3,4a,5,6,7,8-octahydro-10*H*-pyrrolo[1′,2′:2,3]imidazo[1,5-*a*]indol-10-ylidene)-1-phenylethan-1-one (**5d**). Yield 108 mg, 60%. *E*/*Z*-ratio ~4:1. Yellow solid, mp 135–138 °C (hexane/ethyl acetate).

^1^Н NMR (400.1 MHz, CDCl_3_): 7.99–7.97 (m, 2H, H*_o_*, Ph), 7.46 (s, 1H, H-9), 7.44–7.41 (m, 3H, H*_m_*_,*p*_, Ph), 6.00 (s, 1H, =CH), 3.65–3.59 (m, 1H, H-1), 3.53–3.47 (m, 1H, H-1), 2.66–2.59 (m, 4H, H-5, H-8), 2.41–2.28 (m, 2H, H-3), 2.21–2.06 (m, 3H, H-2, C*H*Me_2_), 1.94–1.74 (m, 4H, H-6, H-7), 1.06 (d, *J* = 6.7 Hz, 3H, *Me*CH), 0.41 (d, *J* = 6.0 Hz, 3H, *Me*CH)—*E*-isomer; 7.99–7.97 (m, 2H, H*_o_*, Ph), 7.44–7.41 (m, 3H, H*_m_*_,*p*_, Ph), 6.20 (s, 1H, H-9), 6.16 (s, 1H, =CH), 4.03–3.96 (m, 1H, H-1), 3.41–3.34 (m, 1H, H-1), 2.66–2.59 (m, 4H, H-5, H-8), 2.41–2.28 (m, 2H, H-2), 2.21–2.06 (m, 3H, H-3, C*H*Me_2_), 1.94–1.74 (m, 4H, H-6, H-7), 1.14 (d, *J* = 6.8 Hz, 3H, *Me*CH), 0.41 (d, *J* = 6.0 Hz, 3H, *Me*CH)—*Z*-isomer.

^13^С{^1^H} NMR (100.6 MHz, CDCl_3_): 187.3 (C=O), 156.6 (C-10), 141.8 (C*_i_*, Ph), 131.2 (C-9a), 130.5 (C*_p_*, Ph), 128.7 (C-4a), 128.1 (C*_m_*, Ph), 127.4 (C*_o_*, Ph), 125.3 (C-8a), 110.8 (C-10), 91.5 (=CH), 90.8 (C-3a), 51.9 (C-1), 37.7 (*C*HMe_2_), 32.3 (C-3), 26.7 (C-2), 23.5, 23.4, 23.3, 22.8 (C-5–8), 17.7 (CH*Me*), 15.9 (CH*Me*)—*E*-isomer; 186.0 (C=O), 154.8 (C-10), 141.5 (C*_i_*, Ph), 133.4 (C-9a), 130.6 (C*_p_*, Ph), 128.7 (C*_m_*, Ph), 127.9 (C-4a), 127.5 (C*_o_*, Ph), 125.4 (C-8a), 100.7 (C-9), 93.6 (C-3a), 87.9 (=CH), 53.8 (C-1), 37.8 (*C*HMe_2_), 32.8 (C-3), 26.8 (C-2), 23.7, 23.4, 23.2, 22.7 (C-5–8), 17.7 (CH*Me*), 15.7 (CH*Me*)—*Z*-isomer.

^15^N NMR (40.6 MHz, CDCl_3_): −256.4 (N-11), −194.4 (N-4)—*E*-isomer; −255.1 (N-11), −194.4 (N-4)—*Z*-isomer.

IR (film): *ν* = 1634 (C=O), 1579 (C=C).

MS (EI): *m/z* (%) = 360 (M^+^, 38%), 317 (95), 255 (100), 105 (PhCO^+^, 81), 77 (Ph^+^, 74), 41 (21).

HRMS (ESI-TOF): found 361.2280. Calcd. for [C_24_H_28_N_2_O+H]^+^ 361.2280.

2-(3a-Methyl-1,2,3,3a,5,6,7,8-octahydro-10*H*-pyrrolo[1′,2′:2,3]imidazo[1,5-*a*]indol-10-ylidene)-1-(thiophen-2-yl)ethan-1-one (**5f**). Yield 122 mg, 72%. *E*/*Z*-ratio ~4:1. Orange solid, mp 82–84 °C (hexane).

^1^Н NMR (400.1 MHz, CDCl_3_): 7.64 (dd, *J* = 3.7, 1.0 Hz, 1H, H-3′), 7.45 (dd, *J* = 4.9, 1.0 Hz, 1H, H-5′), 7.44 (s, 1H, H-9), 7.08 (dd, *J* = 4.9, 3.7 Hz, 1H, H-4′), 5.79 (s, 1H, =CH), 3.65–3.59 (m, 1H, H-1), 3.42 (dt, *J* = 10.9, 8.3 Hz, 1H, H-1), 2.72–2.58 (m, 4H, H-5, H-8), 2.36–2.20 (m, 3H, H-2, H-3), 1.96–1.89 (m, 1H, H-3), 1.83–1.74 (m, 4H, H-6, H-7), 1.61 (s, 3H, Me-3a)—*E*-isomer; 7.64 (dd, *J* = 3.7, 1.0 Hz, 1H, H-3′), 7.45 (dd, *J* = 4.9, 1.0 Hz, 1H, H-5′), 7.08 (dd, *J* = 4.9, 3.7 Hz, 1H, H-4′), 6.20 (s, 1H, H-9), 6.09 (s, 1H, =CH), 4.33–4.27 (m, 1H, H-1), 3.25–3.18 (m, 1H, H-1), 2.72–2.58 (m, 4H, H-5, H-8), 2.36–2.20 (m, 3H, H-2, H-3), 1.96–1.89 (m, 1H, H-3), 1.83–1.74 (m, 4H, H-6, H-7), 1.66 (s, 3H, Me-3a)—*Z*-isomer.

^13^С{^1^H} NMR (100.6 MHz, CDCl_3_): 179.6 (C=O), 154.3 (C-10), 148.9 (C-2′), 130.0 (C-5′), 129.7 (C-9a), 128.4 (C-4a), 127.8 (C-3′), 127.6 (C-4′), 125.0 (C-8a), 111.4 (C-9), 89.8 (=CH), 84.9 (C-4a), 48.5 (C-1), 35.2 (C-3), 26.9 (C-2), 25.6 (Me-3a), 23.5, 23.3, 23.1, 22.4 (C-5–8)—*E*-isomer; 178.6 (C=O), 152.8 (C-10), 149.1 (C-2′), 131.9 (C-9a), 130.0 (C-5′), 128.0 (C-4a), 127.6 (C-3′, C-4′), 125.2 (C-8a), 101.4 (C-9), 88.0 (C-3a), 87.1 (=CH), 51.5 (C-1), 36.1 (C-3), 27.0 (C-2), 25.8 (Me-3a), 23.6, 23.3, 23.0, 22.3 (C-5–8)—*Z*-isomer.

^15^N NMR (40.6 MHz, CDCl_3_): −254.4 (N-11), −192.5 (N-4)—*E*-isomer; −246.5 (N-11), −192.4 (N-4)—*Z*-isomer.

IR (film): *ν* = 1618 (C=O), 1535 (C=C).

MS (EI): *m/z* (%) = 338 (M^+^, 21%), 227 (100), 111 (58), 83 (10), 41 (15), 39 (25).

HRMS (ESI-TOF): found 339.1531. Calcd for [C_20_H_22_N_2_OS+H]^+^ 339.1531.

2-(5,6,10,11,12,12a-Hexahydro-8*H*-benzo[*g*]pyrrolo[2′,1′:2,3]imidazo[1,5-*a*]indol-8-yidene)-1-phenylethan-1-one (**6a**). Yield 128 mg, 70%. *E*/*Z*-ratio ~3:1. Yellow solid, mp 156–158 °C (hexane).

^1^Н NMR (400.1 MHz, CDCl_3_): 8.00–7.98 (m, 2H, H*_o_*, Ph), 7.70 (s, 1H, H-7), 7.49–7.42 (m, 3H, H*_m_*_,*p*_, Ph), 7.35–7.32 (m, 1H, H-1), 7.29–7.24 (m, 2H, H-2, H-4), 7.16 (dd, *J* = 7.4, 7.4 Hz, 1H, H-3), 5.99 (s, 1H, =CH), 5.90 (dd, *J* = 8.8, 5.1 Hz, 1H, H-12a), 3.70–3.65 (m, 1H, H-10), 3.52–3.45 (m, 1H, H-10), 2.99–2.93 (m, 2H, H-6), 2.87–2.75 (m, 2H, H-5), 2.72–2.66 (m, 1H, H-12), 2.40–2.31 (m, 2H, H-11), 1.72–1.62 (m, 1H, H-12)—*E*-isomer; 8.00–7.98 (m, 2H, H*_o_*, Ph), 7.44–7.42 (m, 3H, H*_m_*_,*p*_, Ph), 7.35–7.32 (m, 1H, H-1), 7.29–7.24 (m, 2H, H-2, H-4), 7.16 (dd, *J* = 7.4, 7.4 Hz, 1H, H-3), 6.37 (s, 1H, H-7), 6.28 (s, 1H, =CH), 6.05 (dd, *J* = 8.8, 5.1 Hz, 1H, H-12a), 4.40–4.34 (m, 1H, H-10), 3.20 (dt, *J* = 11.6, 8.8 Hz, 1H, H-10), 2.77–2.66 (m, 4H, H-5, H-6), 2.40–2.31 (m, 2H, H-12), 1.85–1.62 (m, 2H, H-11)—*Z*-isomer.

^13^С{^1^H} NMR (100.6 MHz, CDCl_3_): 187.4 (C=O), 154.0 (C-8), 141.5 (C*_i_*, Ph), 136.9 (C-13b), 133.2 (C-7a), 130.8 (C*_p_*, Ph), 128.9 (C-1), 128.5 (C-6a), 128.3 (C-13a), 128.2 (C*_m_*, Ph), 127.5 (C*_o_*, Ph), 127.3 (C-4a), 126.7 (C-4), 126.5 (C-2), 120.0 (C-3), 111.1 (C-7), 90.8 (=CH), 78.7 (C-12a), 48.8 (C-10), 31.3 (C-12), 30.3 (C-5), 26.5 (C-11), 22.2 (C-6)—*E*-isomer; 186.6 (C=O), 152.7 (C-8), 141.2 (C*_i_*, Ph), 136.1 (C-13b), 135.3 (C-7a), 130.9 (C*_p_*, Ph), 128.9 (C-1), 127.9 (C-6a), 127.8 (C-13a), 127.6 (C*_m_*, Ph), 127.5 (C*_o_*, Ph), 127.3 (C-4a), 126.8 (C-4), 126.4 (C-2), 120.1 (C-3), 101.8 (C-7), 88.5 (=CH), 81.2 (C-12a), 51.9 (C-10), 32.2 (C-12), 30.1 (C-5), 26.7 (C-11), 22.4 (C-6)—*Z*-isomer.

^15^N NMR (40.6 MHz, CDCl_3_): −256.7 (N-9), −212.2 (N-13)—*E*-isomer; −258.3 (N-9), −213.0 (N-13)—*Z*-isomer.

IR (film): *ν* = 1629 (C=O), 1578 (C=C).

MS (EI): *m/z* (%) = 366 (M^+^, 21%), 261 (13), 192 (13), 191 (17), 105 (67), 77 (100), 51 (22), 41 (19).

HRMS (ESI-TOF): found 367.18081. Calcd. for [C_25_H_22_N_2_O+H]^+^ 367.18104.

2-(12a-Methyl-5,6,10,11,12,12a-hexahydro-8*H*-benzo[*g*]pyrrolo[2′,1′:2,3]imidazo[1,5-*a*]indol-8-yidene)-1-phenylethan-1-one (**6b**). Yield 84 mg, 44%. *E*/*Z*-ratio ~3:1. Yellow solid, mp 102–106 °C.

^1^Н NMR (400.1 MHz, CDCl_3_): 8.00–7.97 (m, 2H, H*_o_*, Ph), 7.71 (s, 1H, H-7), 7.52–7.42 (m, 4H, H-1, H*_m_*_,*p*_, Ph), 7.28–7.22 (m, 2H, H-2, H-4), 7.15 (dd, *J* = 7.5, 7.5 Hz, 1H, H-3), 5.90 (s, 1H, =CH), 3.70–3.66 (m, 1H, H-10), 3.54 (dt, *J* = 9.3, 9.2 Hz, 1H, H-10), 3.03–2.66 (m, 5H, H-6, H-11, H-12), 2.50–2.46 (m, 2H, H-5), 2.36–2.27 (m, 1H, H-12), 1.62 (s, 3H, Me)—*E*-isomer; 8.00–7.97 (m, 2H, H*_o_*, Ph), 7.52–7.42 (m, 4H, H-1, H*_m_*_,*p*_, Ph), 7.28–7.22 (m, 2H, H-2, H-4), 7.15 (dd, *J* = 7.5, 7.5 Hz, 1H, H-3), 6.37 (s, 1H, H-7), 6.20 (s, 1H, =CH), 4.42 (dd, *J* = 12.0, 8.0 Hz, 1H, H-10), 3.27–3.20 (m, 1H, H-10), 3.03–2.66 (m, 6H, H-6, H-11, H-12), 2.50–2.46 (m, 2H, H-5), 1.70 (s, 3H, Me)—*Z*-isomer.

^13^С{^1^H} NMR (100.6 MHz, CDCl_3_): 187.3 (C=O), 152.7 (C-8), 141.7 (C*_i_*, Ph), 137.2 (C-13b), 134.1 (C-7a), 130.6 (C*_p_*, Ph), 129.5 (C-6a), 129.3 (C-13a), 129.0 (C-1), 128.6 (C-4a), 128.2 (C*_m_*, Ph), 127.5 (C*_o_*, Ph), 126.6 (C-4), 126.3 (C-2), 121.1 (C-3), 111.6 (C-7), 90.1 (=CH), 87.1 (C-12a), 47.4 (C-10), 37.4 (C-12), 30.5 (C-5), 26.6 (C-11), 25.7 (Me), 22.7 (C-6)—*E*-isomer; 186.3 (C=O), 151.4 (C-8), 141.5 (C*_i_*, Ph), 136.5 (C-13b), 136.2 (C-7a), 130.8 (C*_p_*, Ph), 129.6 (C-6a), 128.9 (C-1), 128.8 (C-13a), 128.6 (C-4a), 128.2 (C*_m_*, Ph), 127.6 (C*_o_*, Ph), 126.7 (C-4), 126.2 (C-2), 120.9 (C-3), 102.4 (C-7), 90.5 (=CH), 87.5 (C-12a), 50.6 (C-10), 38.3 (C-12), 30.3 (C-5), 26.7 (C-11), 25.9 (Me), 22.9 (C-6)—*Z*-isomer.

^15^N NMR (40.6 MHz, CDCl_3_): −253.2 (N-9), −200.4 (N-13)—*E*-isomer; −244.4 (N-9), −200.4 (N-13)—*Z*-isomer.

IR (film): *ν* = 1629 (C=O), 1577 (C=C).

MS (EI): *m/z* (%) = 380 (M^+^, 14%), 276 (19), 275 (83), 105 (71), 77 (100), 51 (19), 41 (15).

HRMS (ESI-TOF): found 381.1965. Calcd. for [C_26_H_24_N_2_O+H]^+^ 381.1969.

2-(12a-Ethyl-5,6,10,11,12,12a-hexahydro-8*H*-benzo[*g*]pyrrolo[2′,1′:2,3]imidazo[1,5-*a*]indol-8-yidene)-1-phenylethan-1-one (**6c**). Yield 66 mg, 33 %. *E*/*Z*-ratio ~3:1. Yellow solid, mp 114–118 °C.

^1^Н NMR (400.1 MHz, CDCl_3_): 8.00–7.98 (m, 2H, H*_o_*, Ph), 7.66 (s, 1H, H-7), 7.48–7.42 (m, 4H, H-1, H*_m_*_,*p*_, Ph), 7.28–7.27 (m, 1H, H-4), 7.23 (t, *J* = 7.4 Hz, 1H, H-2), 7.14 (t, *J* = 7.4 Hz, 1H, H-3), 5.94 (s, 1H, =CH), 3.63–3.54 (m, 2H, H-10), 3.00–2.62 (m, 5H, H-6, H-11, H-12), 2.51–2.36 (m, 3H, H-5, H-12), 2.17 (dq, *J* = 14.4, 7.2 Hz, 1H, C*H*_2_Me), 1.88 (dq, *J* = 14.4, 7.2 Hz, 1H, C*H*_2_Me), 0.36 (t, *J* = 7.2 Hz, 3 H, *Me*CH_2_)—*E*-isomer; 8.00–7.98 (m, 2H, H*_o_*, Ph), 7.48–7.42 (m, 4H, H-1, H*_m_*_,*p*_, Ph), 7.28–7.27 (m, 1H, H-4), 7.23 (t, *J* = 7.4 Hz, 1H, H-2), 7.14 (t, *J* = 7.4 Hz, 1H, H-3), 6.34 (s, 1H, H-7), 6.21 (s, 1H, =CH), 4.39–4.34 (m, 1H, H-10), 3.31–3.23 (m, 1H, H-10), 3.00–2.62 (m, 5H, H-6, H-11, H-12), 2.51–2.36 (m, 3H, H-5, H-12), 2.05–1.99 (m, 2H, C*H*_2_Me), 0.39 (t, *J* = 7.2 Hz, 3H, *Me*CH_2_)—*Z*-isomer.

^13^С{^1^H} NMR (100.6 MHz, CDCl_3_): 187.2 (C=O), 153.6 (C-8), 141.8 (C*_i_*, Ph), 137.1 (C-13b), 135.7 (C-7a), 130.6 (C*_p_*, Ph), 129.8 (C-6a), 129.4 (C-13a), 128.9 (C-1), 128.7 (C-4a), 128.1 (C*_m_*, Ph), 127.4 (C*_o_*, Ph), 126.5 (C-4), 126.2 (C-2), 121.2 (C-3), 111.1 (C-7), 90.2 (C-12a), 89.8 (=CH), 47.3 (C-10), 37.0 (C-12), 30.8 (*C*H_2_Me), 30.6 (C-5), 26.5 (C-11), 22.7 (C-6), 6.4 (Me)—*E*-isomer; 186.0 (C=O), 151.9 (C-8), 141.5 (C*_i_*, Ph), 138.0 (C-13b), 136.4 (C-7a), 130.7 (C*_p_*, Ph), 129.6 (C-6a), 128.9 (C-1), 128.8 (C-13a), 128.7 (C-4a), 128.1 (C*_m_*, Ph), 127.5 (C*_o_*, Ph), 126.7 (C-4), 126.1 (C-2), 120.9 (C-3), 101.8 (C-7), 93.5 (C-12a), 87.1 (=CH), 50.5 (C-10), 37.9 (C-12), 31.2 (*C*H_2_Me), 30.4 (C-5), 26.6 (C-11), 22.8 (C-6), 6.4 (Me)—*Z*-isomer.

^15^N NMR (40.6 MHz, CDCl_3_): −258.4 (N-9), −204.8 (N-13)—*E*-isomer; −250.3 (N-9), −204.1 (N-13)—*Z*-isomer.

IR (film): *ν* = 1630 (C=O), 1578 (C=C).

MS (EI): *m/z* (%) = 394 (M^+^, 21%), 379 (13), 289 (100), 105 (48), 77 (77), 51 (11), 41 (15).

HRMS (ESI-TOF): found 395.21224. Calcd. for [C_27_H_26_N_2_O+H]^+^ 395.2123.

2-(12a-Isopropyl-5,6,10,11,12,12a-hexahydro-8*H*-benzo[*g*]pyrrolo[2′,1′:2,3]imidazo[1,5*-a*]indol-8-yidene)-1-phenylethan-1-one (**6d**). Yield 54 mg, 26 %. *E*/*Z*-ratio ~3:1. Yellow solid, mp 125–127 °C.

^1^Н NMR (400.1 MHz, CDCl_3_): 8.02–7.99 (m, 2H, H*_o_*, Ph), 7.66 (s, 1H, H-7), 7.49–7.43 (m, 4H, H*_m_*_,*p*_, Ph, H-1), 7.28–7.22 (m, 2H, H-2, H-4), 7.14 (dd, *J* = 7.4, 7.4 Hz, 1H, H-3), 6.01 (s, 1H, =CH), 3.74–3.61 (m, 2H, H-10), 2.97–2.67 (m, 5H, H-6, H-11, H-12), 2.65–2.53 (m, 1H, H-12), 2.49–2.36 (m, 3H, C*H*Me_2_, H-5), 1.04 (d, *J* = 6.8 Hz, 3H, *Me*CH), 0.26 (d, *J* = 6.5 Hz, 3H, *Me*CH)—*E*-isomer; 8.02–7.99 (m, 2H, H*_o_*, Ph), 7.49–7.43 (m, 4H, H*_m_*_,*p*_, Ph, H-1), 7.28–7.22 (m, 2H, H-2, H-4), 7.14 (dd, *J* = 7.4, 7.4 Hz, 1H, H-3), 6.33 (s, 1H, H-7), 6.25 (s, 1H, =CH), 4.29–4.22 (m, 1H, H-10), 3.40–3.33 (m, 1H, H-10), 3.00–2.36 (m, 9H, H-5; H-6, H-11, H-12, C*H*Me_2_), 1.13 (d, *J* = 6.8 Hz, 3H, *Me*CH), 0.26 (d, *J* = 6.0 Hz, 3H, *Me*CH)—*Z*-isomer.

^13^С{^1^H} NMR (100.6 MHz, CDCl_3_): 187.4 (C=O), 154.8 (C-8), 141.8 (C*_i_*, Ph), 137.3 (C-13b), 135.4 (C-7a), 130.7 (C*_p_*, Ph), 130.0 (C-6a), 129.9 (C-13a), 129.0 (C-4a), 128.9 (C-1), 128.2 (C*_m_*, Ph), 127.5 (C*_o_*, Ph), 126.6 (C-4), 126.2 (C-2), 121.5 (C-3), 111.0 (C-7), 93.0 (C-12a), 91.1 (=CH), 50.1 (C-10), 38.1 (C-12), 35.0 (*C*HMe_2_), 30.7 (C-5), 26.7 (C-11), 22.8 (C-6), 17.8 (Me), 15.7 (Me)—*E*-isomer; 186.1 (C=O), 153.0 (C-8), 141.5 (C*_i_*, Ph), 137.7 (C-13b), 136.6 (C-7a), 130.8 (C*_p_*, Ph), 130.0 (C-6a), 129.4 (C-13a), 129.0 (C-4a), 128.9 (C-1), 128.2 (C*_m_*, Ph), 127.6 (C*_o_*, Ph), 126.7 (C-4), 126.1 (C-2), 121.3 (C-3), 101.6 (C-7), 96.2 (C-12a), 87.7 (=CH), 52.5 (C-10), 38.1 (C-12), 35.6 (*C*HMe_2_), 30.6 (C-5), 26.8 (C-11), 22.9 (C-6), 17.7 (Me), 15.6 (Me)—*Z*-isomer.

^15^N NMR (40.6 MHz, CDCl_3_): −262.8 (N-9), −201.2 (N-13)—*E*-isomer; −254.7 (N-9), −198.9 (N-13)—*Z*-isomer. I

R (film): *ν* = 1632 (C=O), 1579 (C=C).

MS (EI): *m/z* (%) = 408 (M^+^, 46%), 393 (14), 365 (14), 304 (25), 303 (100), 288 (44), 105 (57), 77 (51), 41 (12).

HRMS (ESI-TOF): found 409.22764. Calcd. for [C_28_H_28_N_2_O+H]^+^ 409.22799.

2-(1,2,3,3a,6,7,8,9-Octahydrocyclohepta[4,5]pyrrolo[1,2-*c*]pyrrolo[1,2-*a*]imidazol-11(5*H*)-ylidene)-1-phenylethan-1-one (**7a**). Yield 126 mg, 76 %. *E*/*Z*-ratio ~4:1. Yellow solid, mp 178–180 °C.

^1^Н NMR (400.1 MHz, CDCl_3_) 7.97–7.96 (m, 2H, H*_o_*_,_ Ph), 7.48 (s, 1H, H-10), 7.44–7.37 (m, 3H, H*_m_*_,*p*,_ Ph), 5.91 (s, 1 H, =CH), 5.46 (dd, *J* = 8.9, 5.4 Hz, 1H, H-3a), 3.61–3.56 (m, 1H, H-1), 3.37 (dt, *J* = 9.9, 7.6 Hz, 1H, H-1), 2.71–2.60 (m, 4H, H-5, H-9), 2.40–2.23 (m, 3H, H-2, H-3), 1.86–1.67 (m, 6H, H-6, H-7, H-8), 1.60–1.50 (m, 1H, H-3)—*E*-isomer; 7.97–7.96 (m, 2H, H*_o_*_,_ Ph), 7.44–7.37 (m, 3H, H*_m_*_,*p*_, Ph), 6.21 (s, 1H, H-10), 6.17 (s, 1H, =CH), 5.63 (dd, *J* = 8.6, 5.7 Hz, 1H, H-3a), 4.31–4.29 (m, 1H, H-1), 3.09 (dt, *J* = 11.7, 8.5 Hz, 1H, H-1), 2.71–2.60 (m, 4H, H-5, H-9), 2.40–2.23 (m, 3H, H-2, H-3), 1.86–1.67 (m, 7H, H-3, H-6, H-7, H-8)—*Z*-isomer.

^13^С{^1^H} NMR (100.6 MHz, CDCl_3_) 187.2 (C=O), 155.2 (C-11), 141.7 (C*_i_*, Ph), 132.5 (C-10a), 130.7 (C*_p_*, Ph), 129.6 (C-4a), 128.7 (C-9a), 128.1 (C*_m_*, Ph), 127.5 (C*_o_*, Ph), 113.3 (C-10), 90.1 (=CH), 77.2 (C-3a), 48.4 (C-1), 32.0 (C-3), 31.0, 29.0, 28.9, 28.0, 27.6 (C-6–9), 26.7 (C-2)—*E*-isomer; 186.4 (C=O), 153.8 (C-10), 141.5 (C*_i_*, Ph), 131.6 (C-9a), 130.8 (C*_p_*_,_ Ph), 130.7 (C-4a), 130.0 (C-9a), 128.1 (C*_m_*_,_ Ph), 127.6 (C*_o_*, Ph), 103.7 (C-10), 88.0 (=CH), 79.7 (C-3a), 51.3 (C-1), 31.9 (C-3), 31.6, 29.0, 28.9, 28.0, 27.5 (C-5–9), 26.9 (C-2)—*Z*-isomer.

^15^N NMR (40.5 MHz, CDCl_3_): −265.3 (N-12), −201.9 (N-4)—*E*-isomer; −257.6 (N-12), −201.9 (N-4)—*Z*-isomer.

IR (film): *ν* = 1630 (C=O), 1578 (C=C).

MS (EI): *m/z* (%) = 332 (M^+^, 100%), 304 (11), 303 (13), 227 (51), 105 (52), 77 (68), 51 (12), 41 (17).

HRMS (ESI): found 333.1969. Calcd. for [C_22_H_24_N_2_O +H]^+^ 333.1967.

2-(3a-Methyl-1,2,3,3a,6,7,8,9-octahydrocyclohepta[4,5]pyrrolo[1,2-*c*]pyrrolo[1,2-*a*]imidazol-11(5*H*)-ylidene)-1-phenylethan-1-one (**7b**). Yield 114 mg, 66 %. *E*/*Z*-ratio ~4:1. Yellow solid, mp 128–130 °C.

^1^Н NMR (400.1 MHz, CDCl_3_): 7.97–7.95 (m, 2H, H*_o_*, Ph), 7.50 (s, 1H, H-10), 7.43–7.40 (m, 3H, H*_m_*_,*p*_, Ph), 5.85 (s, 1H, =CH), 3.64–3.58 (m, 1H, H-1), 3.39 (dd, *J* = 18.7, 8.5 Hz, 1H, H-1), 2.78–2.76 (m, 2H, H-5), 2.67–2.65 (m, 2H, H-9), 2.33–2.17 (m, 3H, H-2, H-3), 1.88–1.70 (m, 7H, H-3, H-6, H-7, H-8), 1.58 (s, 3H, Me-3a)—*E*-isomer; 7.97–7.95 (m, 2H, H*_o_*, Ph), 7.43–7.40 (m, 3H, H*_m_*_,*p*_, Ph), 6.19 (s, 1H, H-10), 6.12 (s, 1H, =CH), 4.35–4.30 (m, 1H, H-1), 3.17–3.10 (m, 1H, H-1), 2.78–2.76 (m, 2H, H-5), 2.62–2.59 (m, 2H, H-9), 2.33–2.17 (m, 3H, H-2, H-3), 1.88–1.70 (m, 7H, H-3, H-6, H-7, H-8), 1.65 (s, 3H, Me-3a)—*Z*-isomer.

^13^С{^1^H} NMR (100.6 MHz, CDCl_3_): 187.2 (C=O), 154.2 (C-11), 141.8 (C*_i_*, Ph), 132.6 (C-10a), 130.4 (C-4a), 130.3 (C*_p_*, Ph), 128.1 (C*_m_*, Ph), 127.9 (C-9a), 127.4 (C*_o_*, Ph), 113.2 (C-10), 89.9 (=CH), 85.0 (C-3a), 48.2 (C-1), 35.5 (C-3), 32.3, 28.9, 28.8, 27.8, 27.7 (C-5–9), 26.9 (Me-3a), 26.8 (C-2)—*E*-isomer; 186.2 (C=O), 152.8 (C-11), 141.6 (C*_i_*, Ph), 131.6 (C-10a), 130.6 (C*_p_*, Ph), 130.6 (C-4a), 130.0 (C-9a), 128.1 (C*_m_*, Ph), 127.5 (C*_o_*, Ph), 103.5 (C-10), 88.1 (C-3a), 87.4 (=CH), 51.0 (C-1), 36.4 (C-3), 32.2, 29.0, 28.8, 27.7, 27.5 (C-5–9), 27.0 (Me-3a, C-2)—*Z*-isomer.

^15^N NMR (40.6 MHz, CDCl_3_): −254.7 (N-12), −190.1 (N-4)—*E*-isomer; −245.2 (N-12), −190.1 (N-4)—*Z*-isomer.

IR (film): *ν* = 1631 (C=O), 1578 (C=C).

MS (EI): *m/z* (%) = 346 (M^+^, 23%), 345 (15), 241 (100), 105 (55), 77 (66), 51 (11), 41 (15).

HRMS (ESI-TOF): found 347.21262. Calcd. for [C_23_H_26_N_2_O+H]^+^ 347.21234.

2-(3a-Ethyl-1,2,3,3a,6,7,8,9-octahydrocyclohepta[4,5]pyrrolo[1,2-*c*]pyrrolo[1,2-*a*]imidazol-11(5*H*)-ylidene)-1-phenylethan-1-one (**7c**). Yield 61 mg, 34 %. *E*/*Z*-ratio ~4:1. Yellow solid, mp 132–134 °C.

^1^Н NMR (400.1 MHz, CDCl_3_): 7.97–7.95 (m, 2H, H*_o_*, Ph), 7.46 (s, 1H, H-10), 7.43–7.01 (m, 3H, H*_m_*_,*p*,_ Ph), 5.89 (s, 1H, =CH), 3.59–3.53 (m, 1H, H-1), 3.48–3.42 (m, 1H, H-1), 2.75–2.65 (m, 4H, H-5, H-9), 2.33–2.21 (m, 3H, H-2, H-3), 1.98–1.81 (m, 4H, H-7, C*H*_2_Me), 1.69–1.67 (m, 5H, H-3, H-6, H-8), 0.47 (t, *J* = 7.2 Hz, 3H, *Me*CH_2_)—*E-*isomer; 7.97–7.95 (m, 2H, H*_o_*, Ph), 7.43–7.40 (m, 3H, H*_m_*_,*p*_, Ph), 6.17 (s, 1H, H-10), 6.12 (s, 1H, =CH), 4.26–4.20 (m, 1H, H-1), 3.24–3.17 (m, 1H, H-1), 2.75–2.64 (m, 4H, H-5, H-9), 2.33–2.21 (m, 3H, H-2, H-3), 2.09–1.99 (m, 5H, H-3, H-7, C*H*_2_Me), 1.69–1.67 (m, 4H, H-6, H-8), 0.47 (t, *J* = 7.2 Hz, 3H, *Me*CH_2_)—*Z*-isomer.

^13^С{^1^H} NMR (100.6 MHz, CDCl_3_): 187.2 (C=O), 155.1 (C-11), 141.9 (C*_i_*, Ph), 133.0 (C-10a), 130.4 (C-5a, C*_p_*, Ph), 129.4 (C-9a), 128.1 (C*_m_*, Ph), 127.4 (C*_o_*, Ph), 112.8 (C-10), 89.7 (=CH), 88.2 (C-3a), 48.3 (C-1), 35.1 (*C*H_2_Me), 32.4 (C-2), 32.2, 29.0×2, 28.0, 27.6 (C-5–9), 26.8 (C-3), 6.6 (CH_2_*Me*)—*E*-isomer; 186.0 (C=O), 153.5 (C-11), 141.6 (C*_i_*, Ph), 132.0 (C-10a), 131.6 (C-4a), 130.7 (C*_p_*, Ph), 130.6 (C-9a), 128.1 (C*_m_*, Ph), 127.5 (C*_o_*_,_ Ph), 103.0 (C-10), 91.2 (C-3a), 87.1 (=CH), 51.0 (C-1), 35.9 (*C*H_2_Me), 32.5 (C-3), 32.3, 29.8, 29.1, 27.8, 27.5 (C-5–9), 26.9 (C-2), 6.6 (CH_2_*Me*)—*Z*-isomer.

^15^N NMR (40.5 MHz, CDCl_3_): −260.6 (N-12), −194.6 (N-4)—*E*-isomer; −251.0 (N-12), −194.6 (N-4)—*Z*-isomer.

IR (film): *ν* = 1631 (C=O), 1578 (C=C).

MS (EI): *m/z* (%) = 360 (M^+^, 19%), 255 (99), 105 (64), 77 (100), 51 (22), 41 (37).

HRMS (ESI): found 361.2279. Calcd. for [C_24_H_28_N_2_O+H]^+^ 361.2280.

2-(3a-Isopropyl-1,2,3,3a,6,7,8,9-octahydrocyclohepta[4,5]pyrrolo[1,2-*c*]pyrrolo[1,2-*a*]imidazol-11(5*H*)-ylidene)-1-phenylethan-1-one (**7d**). Yield 59 mg, 32 %. *E*/*Z*-ratio ~4:1. Yellow solid, mp 140–143 °C (hexane).

^1^Н NMR (400.1 MHz, CDCl_3_): 7.98–7.97 (m, 2H, H*_o_*, Ph), 7.46 (s, 1H, H-10), 7.43–7.40 (m, 3H, H*_m_*_,*p*_, Ph), 5.97 (s, 1H, =CH), 3.60–3.53 (m, 1H, H-1), 3.51–3.47 (m, 1H, H-1), 2.74–2.64 (m, 4H, H-5, H-9), 2.42–2.37 (m, 1H, H-3), 2.24–2.21 (m, 3H, H-2, C*H*Me), 2.14–2.06 (m, 1H, H-3), 1.84–1.67 (m, 6H, H-6, H-7, H-8), 1.06 (d, *J* = 6.7 Hz, 3H, *Me*CH), 0.37 (d, *J* = 5.7 Hz, 3H, *Me*CH)—*E*-isomer; 7.98–7.97 (m, 2H, H*_o_*, Ph), 7.43–7.40 (m, 3H, H*_m_*_,*p*_, Ph), 6.17 (s, 2H, H-10, =CH), 4.07–4.01 (m, 1H, H-1), 3.37–3.31 (m, 1H, H-1), 2.74–2.58 (m, 4H, H-5, H-9), 2.42–2.37 (m, 1H, H-3), 2.14–2.06 (m, 4H, H-2, H-3, C*H*Me), 1.84–1.67 (m, 6H, H-6, H-7, H-8), 1.13 (d, *J* = 6.6 Hz, 3H, *Me*CH), 0.37 (d, *J* = 5.7 Hz, 3H, *Me*CH)—*Z*-isomer.

^13^С{^1^H} NMR (100.6 MHz, CDCl_3_): 187.4 (C=O), 156.3 (C-11), 141.9 (C*_i_*, Ph), 133.3 (C-10a), 131.1 (C-4a), 130.5 (C*_p_*, Ph), 129.3 (C-9a), 128.1 (C*_m_*, Ph), 127.5 (C*_o_*, Ph), 112.8 (C-10), 91.1 (=CH), 91.0 (C-3a), 51.4 (C-1), 39.2 (C-3), 32.9 (*C*HMe_2_), 32.5, 29.0 (2C), 28.0, 27.9 (C-5–9), 26.8 (C-2), 17.8 (CH*Me*), 15.8 (CH*Me*)—*E*-isomer; 186.0 (C=O), 154.6 (C-11), 141.6 (C*_i_*, Ph), 132.3 (C-10a), 131.4 (C-4a), 131.1 (C-9a), 130.6 (C*_p_*, Ph), 128.1 (C*_m_*, Ph), 127.5 (C*_o_*, Ph), 102.9 (C-10), 93.9 (C-3a), 87.7 (=CH), 53.4 (C-1), 39.2 (C-3), 33.4 (*C*HMe_2_), 32.4, 29.0 (2C), 27.9, 27.7 (C-5–9), 26.9 (C-2), 17.8 (CH*Me*), 15.7 (CH*Me*)—*Z*-isomer.

^15^N NMR (40.5 MHz, CDCl_3_): −263.5 (N-12), −189.4 (N-4)—*E*-isomer; −255.4 (N-12), −190.9 (N-4)—*Z*-isomer.

IR (film): *ν* = 1632 (C=O), 1579 (C=C).

MS (EI): *m/z* (%) = 374 (M^+^, 50%), 359 (29), 331 (78), 269 (90), 254 (31), 105 (91), 77 (100), 51 (13), 43 (27), 41 (41).

HRMS (ESI): found 409.22764. Calcd. for [C_25_H_30_N_2_O +H]^+^ 409.22798.

## 4. Conclusions

In conclusion, a one-pot chemo- and regioselective synthesis of pyrrolo[1′,2′:2,3]imidazo[1,5-*a*]indoles and cyclohepta[4,5]pyrrolo[1,2-*c*]pyrrolo[1,2-*a*]imidazoles functionalized with acylethenyl groups in up to 81% yields via mild (70 °C) catalyst-free [3+2] annulation of acylethynylcycloalkanepyrroles with Δ^1^-pyrrolines has been developed. This synthesis represents a cardinal, hardly predictable extension of our recently found methodology for the construction of nitrogen-fused heterocycles via [3+2] annulation of alkynones substituted by pyrrolic moieties with cyclic imines, which opens a straightforward route to biochemically related functionalized polycondensed heterocyclic systems, a prospective platform for drug discovery. The physical-chemical characteristics of compound **6a**, which are crucial for TADF emitters in OLEDs, are supportive in favor of the application of this and similar compounds in optoelectronics.

## Data Availability

The data presented in this study are available in the Appendix A.

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
