# Peer review of "Contributing to Biochemistry and Optoelectronics: Pyrrolo[1′,2′:2,3]imidazo[1,5-a]indoles and Cyclohepta[4,5]pyrrolo[1,2-c]pyrrolo[1,2-a]imidazoles via [3+2] Annulation of Acylethynylcycloalka[b]pyrroles with Δ1-Pyrrolines"

_ijms, 2023, doi:10.3390/ijms24043404_

Round 1

Reviewer 1 Report

The manuscript entitled "Contributing to biochemistry and optoelectronics: Pyr-rolo[1',2':2,3]imidazo[1,5-a]indoles and cyclohep-ta[4,5]pyrrolo[1,2-c] pyrrolo[1,2-a]imidazoles via [3+2] annulation of acylethynylcycloalka[b]pyrroles with Δ1-pyrrolines" by Oparina et al. brings us many novelties regarding methods of synthesizing new pyrrolo[1,2]imidazoles. All compounds have been fully characterized, and their synthesis is impeccably described. Despite this, the manuscript focuses very much on the characterization of the compounds and not on their potential application.

So:

1. For consultation, a supplementary material file must be attached to the manuscript with all the NMR spectra, mass spectra, etc... These data are crucial for the reader and the reviewer. In addition, these spectra must reflect the purity of the synthesized compounds.

2. The article's introduction focuses on the biomedical/biochemical applications of compounds with the same pyrrole nucleus. However, the rest of the manuscript focuses on the characterization and optoelectronics part. Why do they focus so much on the biological part if then the application for which they study has nothing to do with biology? It doesn't seem very clear.

3. Regarding the optoelectronic part, all the data presented in tables 5 and 6 should be described in the material and methods part as they were obtained. This information must be added in the material and methods section to be reproduced by any reader without difficulties.

4. Does the data in Table 6 objectively state that the synthesized compounds can potentially be used as high-performance emitters in OLEDs? Unfortunately, the authors cannot confirm the usefulness of the prepared compounds, either in the optoelectronic branch or in the biochemical component, making the manuscript appealing only because many compounds were prepared.

5. Several compounds were prepared, but only 6a was studied in detail. The reader needs to understand why all the rest were left out.

Given these repairs to the manuscript, it may be accepted after major revisions.

Author Response

Reviewer 1 writes:

  1. For consultation, a supplementary material file must be attached to the manuscript with all the NMR spectra, mass spectra, etc... These data are crucial for the reader and the reviewer. In addition, these spectra must reflect the purity of the synthesized compounds.

The recommendation is accepted. The revised version of supplementary materials now includes the copies of the 1H, 13C NMR spectra for all synthesized compounds, the 2D spectra (NOESY, COSY, HSQC, HMBC) for 5d, 6a and 7b, high-resolution mass spectra for all synthesized compounds, and UV spectra for 6a.

  1. The article's introduction focuses on the biomedical/biochemical applications of compounds with the same pyrrole nucleus. However, the rest of the manuscript focuses on the characterization and optoelectronics part. Why do they focus so much on the biological part if then the application for which they study has nothing to do with biology? It doesn't seem very clear.

In the introduction, the focus on “biomedical/biochemical applications” of the compounds has been made in order to draw attention of specialists in this area to the synthesized fused pyrroloimidazoles attract, which are prospective building blocks for drug discovery. At the same time, similar fused heterocycles find applications also in other fields, particularly in optoelectronics.

Correspondingly, the introduction has been re-edited and the following sentence has been included into the manuscript.

Such compounds act as blue-emitting luminophores [13], TADF (thermally activated delayed fluorescence) emitters [14], and the compounds with cell imaging properties [15].

  1. Regarding the optoelectronic part, all the data presented in tables 5 and 6 should be described in the material and methods part as they were obtained. This information must be added in the material and methods section to be reproduced by any reader without difficulties.

Done as recommended. These details are included in the material and methods section.

All the solvents employed for the spectroscopic measurements were of UV spectroscopic grade (Merck). For fluorescence measurements dilute solutions with an absorbance of less than 0.1 (at 1 cm optical path length) at the absorption maximum were used. The fluorescence measurements were performed with a 90° standard geometry. The fluorescence quantum yields (ΦF) of 6a was evaluated relative to anthracene (FF = 0.27 in EtOH as the reference and was corrected from the dependence of the refractive index of the solvent [Photoluminescence of SolutionsC. A. Parker. Elsevier Publishing Co., Amsterdam-London-New York 1968]). The temperature for fluorescence measurements was kept constant at 298 K. An analysis of the nature of the transitions, fraction of electron charge transferred (qCT), distance of charge transfer (DCT) and dipole moment variation at excitation (Dm)was carried out using Multiwfn 3.8. software [Lu T, Chen FW. Multiwfn: a multifunctional wavefunction analyzer. J Comput Chem, 2012, 33: 580–592].

  1. Does the data in Table 6 objectively state that the synthesized compounds can potentially be used as high-performance emitters in OLEDs? Unfortunately, the authors cannot confirm the usefulness of the prepared compounds, either in the optoelectronic branch or in the biochemical component, making the manuscript appealing only because many compounds were prepared.

In our opinion, physical-chemical characteristics of compound 6a, crucial for emitters in OLEDs, evaluated quantum-chemically, are supportive in favor of the application of this and similar compounds in optoelectronics.

Consequently, the following sentence has been included into the conclusion:

The physical-chemical characteristics of compound 6a, which are crucial for emitters in OLEDs, are supportive in favor of the application of this and similar compounds in optoelectronics.

  1. Several compounds were prepared, but only 6a was studied in detail. The reader needs to understand why all the rest were left out.

It is clear that optoelectronics-related properties of benzo[g]pyrroloimidazoindoles 6a-d are associated with the benzo[g]indole fragment in their structure as most conjugated and enriched with p-electrons. Since compounds 6a-d differ from each other only by alkyl substituent, their photophysical properties should not be noticeably different. This is why we have measured the relevant spectral characteristics only for one of them (6а).

This paragraph has been included into the manuscript.

Reviewer 2 Report

Manuscript entitled ''  Contributing to biochemistry and optoelectronics: Pyr-2 rolo[1’,2’:2,3]imidazo[1,5-a]indoles and cyclohep-3 ta[4,5]pyrrolo[1,2-c]pyrrolo[1,2-a]imidazoles via [3+2] annula-4 tion of acylethynylcycloalka[b]pyrroles with Δ1-pyrrolines'' by Trofimov et al. deals with the synthesis of novel nitrogen heterocycles and evaluation of their photophysical properties. 

 Article is well written and easy to follow. Reference sections gives a good literature overview and it is appropriate. All new compounds are fully spectroscopically characterized. There is a quality in this paper and it is of general interest to the readers of IJMS, particularly the ones dealing with organic synthesis, heterocyclic chemistry and structural chemistry. Hence, I would suggest that manuscript should be accepted for publication to IJMS with minor changes:

 1) Abstract, first sentence is not clear and should be reformulated.

2) The authors should consider the provision of some experimental evidence of proposed reaction mechanism.

 Some minor points:

  Page 1, Line 30, ''  cyclocondencation '' should be replaced by ''

 cyclocondensation  ''

Page 2, Line 58, '' completed for 24   '' should be replaced by '' completed within 24  ''

Page 3, Line 83, Scheme 1 caption  should be modified by addition of  ''Proposed reaction mechanism of ...  ''

Page 7, Line 217, '' multiplicates   '' should be replaced by ''multiplets   ''

Author Response

  1. Abstract, first sentence is not clear and should be reformulated.

The sentence was corrected as follows:

The available pyrrolylalkynones, having tetrahydroindolyl-, cycloalkanopyrrolyl-, and dihydrobenzo[g]indolyl moieties, acylethynylcycloalka[b]pyrroles, are readily annulated with Δ1-pyrrolines (MeCN/THF, 70 oC, 8 h) to afford a series of novel pyrrolo[1’,2’:2,3]imidazo[1,5-a]indoles and cyclohepta[4,5]pyrrolo[1,2-c]pyrrolo[1,2-a]imidazoles functionalized with acylethenyl group in up to 81% yield.

  1. The authors should consider the provision of some experimental evidence of proposed reaction mechanism.

In a number of publications [Trofimov, B.A.; Belyaeva, K.V. Zwitterionic adducts of N-heterocycles to electrophilic acetylenes as a master key to diversity and complexity of fundamental nitrogen heterocycles. Tetrahedron 2020, 61, 151991 and references cite therein] concerning the reactions of nitrogen nucleophiles (pyridines, quinolones, indolizines, etc.) with activated alkynes, the reversible formation of 1,3(4)-dipolar complexes (zwitterions) as key intermediates was proved to occur. As follows from Scheme 2, this step is implied to be a major one in the reaction studied and therefore its mechanism does not likely differ much in other details from the commonly accepted ones.

This sentence has been included into manuscript.

All the grammar mistakes indicated by the reviewer (pages 1, 2, 7) have been corrected. Scheme 1 caption was changed to: Proposed mechanism of Δ1-pyrroline-catalyzed reaction of acylethynyltetrahydroindole 1 with MeOH.

Reviewer 3 Report

The article is well organized and provides good information on synthesis process. 

1) Please edit the writing/gramma throughout the article. Sometimes it is difficult to follow the sentences. 

2) In Table 1, why there are two numbers for 5b yields in entries 5 and 6, i.e. 85 (80) and 92 (81)? You mentioned yields of 80- 81% in the article. Please specify what do 85 and 92 represent in the table. 

3) Have you tried different solvent ratios of MeCN/THF during optimization? If so, it could be helpful to include the results in the article. 

Author Response

  1. Please edit the writing/gramma throughout the article. Sometimes it is difficult to follow the sentences. 

The language has been checked and corrected.

  1. In Table 1, why there are two numbers for 5b yields in entries 5 and 6, i.e. 85 (80) and 92 (81)? You mentioned yields of 80- 81% in the article. Please specify what do 85 and 92 represent in the table. 

In Table 1, the yields of adduct 5a, were estimated from 1Н NMR spectra (2, 3,5,6-tetramethylbenzene (durene) as the internal standard). In entries 5 and 6 these are 85 and 92%, respectively. The isolated yields of this compound are 80 and 81% (given in parentheses). The corresponding explanation is given in footnote under the Table: b Determined by 1H NMR with 2,3,5,6-tetramethylbenzene (durene) as the internal standard (isolated yield in parentheses).

  1. Have you tried different solvent ratios of MeCN/THF during optimization? If so, it could be helpful to include the results in the article. 

Preliminarily, we have found that the rate of this reaction dropped on going from MeCN to THF. Because tetrahydroindole 1 is scarcely soluble in MeCN, THF was used as a co-solvent and only 1:1 ratio of MeCN/THF provided homogenic reaction mixture. Therefore, other solvent ratios were not employed.

Consequently, the following sentenses has been included to the manuscript:

Only 1:1 ratio of MeCN/THF provided homogenic reaction mixture. Therefore, other solvent ratios were not employed.

Round 2

Reviewer 1 Report

My questions and suggestions were successfully addressed: the supplementary material contains all the data I requested, and several changes were made to the manuscript to improve it. Even so, the manuscript has several spelling errors that must be considered during the proofing process. As a suggestion, Figure 1 should be replaced or completed with similar compounds used in the optoelectronics area since this is the focus of this work.

Therefore, the manuscript should be accepted for publication.